# National Park or Cultural Landscape Preservation? What the Soil Seed Bank Reveals for Plant Diversity Conservation

Tim Drissen [1], Christopher Faust [1], Julia T. Treitler [1], Robin Stadtmann [2], Stefan Zerbe [3] and Jasmin Mantilla-Contreras [1,4,*]

1. Department of Biology, University of Hildesheim, Universitätsplatz 1, 31141 Hildesheim, Germany
2. Department of Soil Protection and Soil Survey, State Authority for Mining, Energy and Geology, Stilleweg 2, 30655 Hanover, Germany
3. Faculty for Science and Technology, Free University of Bozen-Bolzano, Piazza Università 5, 39100 Bolzano, Italy
4. Biological Station Siegen-Wittgenstein, In d. Zitzenbach 2, 57223 Kreuztal, Germany
* Correspondence: mantilla@uni-hildesheim.de (J.M.-C.)

**Abstract:** National parks play an important role in the conservation of biodiversity, mainly excluding human influence following the IUCN approach. However, in Europe, they are often characterized by a high percentage of traditional cultural landscape elements, which require active management. This calls into question whether the national park protection strategy is always appropriate. Here, we follow this question by taking the soil seed bank of various habitats of the Asinara National Park (Sardinia, Italy) as an example. Asinara is a suitable model region, as the island mainly consists of traditional cultural landscape elements, but the main conservation goals include afforestation plans and nature development promotion, which creates a trade-off between the conservation of forest vs. cultural landscapes. We investigated the soil seed bank, standing vegetation, and environmental factors in different cultural and natural habitats. Since the highest species richness and diversity were revealed for cultural vegetation units, they need to be of primary concern regarding the preservation of the island's phytodiversity. Given the main objective of the conservation of biodiversity in the Asinara National Park, we conclude that a biosphere reserve with an adapted sustainable land-use management might be more suitable than a national park to account for both natural and cultural landscape preservation. This conclusion applies to many other European national parks.

**Keywords:** biosphere reserve; conservation strategy; cultural indicators; ecosystem restoration; grazing; IUCN; land-use management; plant species richness; protected area; succession

## 1. Introduction

Despite numerous efforts regarding biodiversity conservation, its loss continues to be a global problem of high significance for humankind, with far-reaching ecological and socio-economic consequences [1,2]. The primary objective of biodiversity protection targets species and communities that require relatively large areas of undisturbed habitats by supporting the underlying ecological structure and natural environmental processes [3]. One main tool for biodiversity conservation is, therefore, the establishment of large protected areas (PAs). The IUCN has globally classified PAs into six categories (Table 1), with a strict nature reserve having the highest protection status, without any further human land-use impact, except for research [4,5]. National parks are in the second highest protection category and are defined by the IUCN as 'large natural or near-natural areas to protect large-scale ecological processes, along with the complement of species and ecosystems characteristic of the area' [3].

**Table 1.** Global categories of protected areas, their objectives, and management [3,5–10]. Based on shared objectives, each IUCN category comprises different approaches. Categories designated under international conventions are listed separately.

| Category | Description | Primary Objective | Applied Management |
|---|---|---|---|
| Strict nature reserve (IUCN Ia) | Area set aside to protect biodiversity with strictly limited human impact; potential reference area for research/monitoring. | To conserve outstanding ecosystems, species, or geodiversity, which were formed mostly by non-human forces and are sensitive to degradation. | Limited human visitation; continuous management to maintain fragments of ecosystems or habitats; restoration through natural processes or time-limited interventions. |
| Wilderness area (IUCN Ib) | Large unmodified or slightly modified area protected and managed to preserve its natural condition and character without significant human habitation. | To protect long-term ecological integrity of natural undisturbed areas with their natural processes for future generations. | Limited human visitation for self-reliant travel; restoring cultural landscapes to near-natural conditions. |
| National Park (IUCN II) | Large natural or near-natural area set aside to protect large-scale ecological processes, characteristic species, and ecosystems, providing a foundation for environmentally and culturally compatible, spiritual, scientific, educational, recreational, and visitor opportunities. | To protect natural biodiversity with its underlying ecological structure and supporting environmental processes; to promote education and recreation. | Internal zoning for controlling human impact in core areas; measures to combine ecosystem protection with recreation; establishing infrastructure for visitors. |
| Natural monument or feature (IUCN III) | Area set aside to protect a specific natural monument, generally quite small and often with high visitor value. | To conserve specific outstanding natural sites with spiritual and/or cultural values; to protect their associated biodiversity and habitats. | Maintaining a natural feature in otherwise cultural or fragmented landscapes; encouraging visitors sometimes in large numbers. |
| Habitat/species management area (IUCN IV) | Area for the protection of species or habitats as fragments of ecosystems. | To maintain, conserve and restore species and habitats; to enable scientific research, but generally as a secondary objective. | Active management to maintain target species and natural or semi-natural habitats through traditional management; development of public education and regular contact of residents with nature. |
| Protected landscape/seascape (IUCN V) | Area protecting a distinct cultural landscape with significant ecological, biological, cultural, and scenic value. | To protect and sustain important landscapes/seascapes and the associated nature created by interactions with humans through traditional management, safeguarding the integrity of the interactions. | Continuing human intervention to maintain the qualities of cultural landscapes, including biodiversity, through traditional management. |
| Protected area with sustainable use of natural resources (IUCN VI) | Area that conserves ecosystems and habitats together with cultural values and traditional natural resource management. | To protect natural ecosystems and use natural resources sustainably when conservation and sustainable use can be mutually beneficial. | Promoting sustainable use of natural resources and environmental products; supporting sustainable livelihoods; facilitating recreation and small-scale tourism. |

**Table 1.** *Cont.*

| Category | Description | Primary Objective | Applied Management |
|---|---|---|---|
| UNESCO Biosphere Reserve | Areas of representative environments for the conservation and management of biological/cultural diversity and economic/social development based on local community efforts and scientific knowledge. | To conserve a representative sample of major ecosystems; to integrate local communities within the biosphere; to combine conservation research, education, training, and monitoring. | Integrating protected core areas with the surrounding lands and uses with differing management intensities at ecosystem level; improving the overall relationship between people and their environment. |
| UNESCO World Heritage Site | Globally important cultural/natural heritage with an outstanding universal value (history, art, science, geology, ecology, biology). | To identify and conserve natural and cultural sites of outstanding universal value; to transmit heritage to future generations. | Integrating heritage protection into comprehensive planning programs; developing scientific/technical studies and research countering threats. |
| UNESCO Global Geopark | Single, unified geographical areas, where sites and landscapes of international geological significance are managed with a holistic concept of protection, education, and sustainable development. | To combine conservation with sustainable development involving local communities; to enhance awareness of key issues facing society (sustainable resource use, climate change, risks of natural disasters). | Creation of local enterprises, new jobs, and high-quality training courses while protecting geological resources of the area; strengthening people's identification with the area. |
| Site of the Ramsar Convention on Wetlands | Internationally important wetlands. | To conserve, sustainably use, and effectively manage wetlands and resources; to cooperate internationally on transboundary wetlands and shared species. | Water, habitat, and species management, creation of zones, and management of the multiple values of the site. |
| Globally Important Agricultural Heritage Systems—'GIAHS' | Globally important site in terms of supporting food and livelihood security, biodiversity, indigenous knowledge systems, and adapted technologies, culture, and outstanding landscapes. | To conserve GIAHS and their associated landscapes, natural resources, agro-biodiversity, and knowledge; to enhance the benefits for local populations; to enable policies to support conservation, resource allocation, and labor. | Promoting 'cultural ecology' (ecotourism, cultural identity products, local gastronomy); empowering rural communities through participatory methods; promoting research and development of rural services for local populations. |

For the recognition as a national park, the IUCN requires that nature must be left without further human land-use impact on at least 75% of the area. However, in Europe, this requirement constitutes a problem, as, here, the landscape has been influenced by a history of land use lasting thousands of years [11] which resulted in unique cultural habitats [12], often with a high value for biodiversity and nature conservation [13]. Thus, many national parks are historically developed cultural landscapes and do not present pristine wilderness, and the fundamental concept of protecting large-scale ecological processes within large natural or near-natural areas is likely to be disregarded.

This is particularly true for the Mediterranean region [14–16], where a long history of land-use practices such as deforestation, animal husbandry, and agriculture resulted in heterogeneous landscapes associated with the region's high biodiversity [17–20]. Today, secondary or semi-natural communities such as maquis, garrigue, and grassland character-ize the Mediterranean landscape, and floristic diversity largely depends on the influence of regular disturbance factors such as cutting, fire, or grazing [18]. Cessation of land-use practices and human-induced disturbances would finally end up in ecosystems dominated by shrubs and evergreen forests, associated with lower diversity [21]. Consequently, the unique biodiversity and landscape structure might be at risk if the IUCN national park con-

cept focusing on undisturbed natural ecosystems and processes (without human influence) is strictly enforced in areas with prevailing cultural features. In those cases, the designation as a national park should be questioned in the first place. However, implementation of national parks within the national law of a respective country can vary substantially from the IUCN requirements [5], including different management intensities, while nature development through non-intervention often remains the fundamental idea. This leads to various possible protection goals and trade-offs (e.g., naturalness vs. cultural landscape). If various protection goals coexist, land-use management should carefully balance the benefits of different vegetation types for the eco-social development of a given region. In addition, nature and biodiversity protection have been proven to be difficult or even impossible in several areas without taking human dimensions into account [22–24]. Accordingly, updated nature conservation concepts have been developed in the past decades, which also include humans, land use, and participatory approaches. This has been implemented with the protection concept of the UNESCO biosphere reserve. Here, the sustainable development of land-use types and the preservation of near-natural core zones or natural succession coexist, aiming at the maintenance of heterogeneously structured landscapes and, therewith, the preservation of biodiversity [25]. Accordingly, 'classic nature conservation' in the narrow sense has been broadened towards an interdisciplinary and transdisciplinary approach, which aims at a sustainable co-existence of humans and nature within a cultural landscape.

Regarding area management and restoration planning of culturally evolved landscapes, over the last decades, there has been a growing interest in the role of seed banks and their relationship to vegetation dynamics and post-disturbance regeneration, especially in Mediterranean-type ecosystems [26–28]. Often seed bank and aboveground vegetation are closely related over space and time, making the seed bank fundamentally important in succession dynamics [28], especially regarding the abandonment of land-use practices [26,27]. Most seed-bank-related studies, however, focus on agricultural rather than semi-natural habitats, although, the latter often have a high value for nature conservation [29,30].

In Italy, few protected areas comprise coastal landscapes [31], making the protection of island ecosystems especially important in the central Mediterranean region. Therefore, our study focuses on the semi-natural Mediterranean ecosystem of Asinara Island (Sardinia, Italy), which is especially of interest in the context of the cessation of traditional land-use practices. Due to a changing history of land use, today, only 0.7% of the island's area consists of evergreen forest; the rest is mostly dominated by secondary shrub vegetation and grassland. However, the island provides a habitat for 20% of the endangered plant species on Sardinia [32]. The conditions of Asinara provide a unique opportunity since the island has a long history of human influence, and it mainly consists of traditional cultural landscape elements, which are defined by Zerbe [33] as those elements which have not yet undergone land-use intensification. Generally, they have been practiced for a long time, are low-input land-use systems, and often are characterized by high biodiversity. Asinara is protected as a national park, but conservation goals include afforestation plans (increasing forest vegetation up to 30% of the island's area), nature development promotion but also the maintenance of diverse cultural habitats. We take the Asinara National Park as an example of the above-mentioned potential trade-off between the protection of natural and cultural habitats and discuss the conservation strategy by taking the soil seed bank as an indicator for the natural future development. Therefore, we focused on the following questions:

1. What are the seed bank characteristics compared to the standing vegetation in a strongly anthropogenic altered semi-natural island landscape?
2. Which environmental factors determine differences in seed bank composition and diversity?
3. What does the seed bank indicate regarding the preservation and sustainable development of plant diversity and the pursued nature conservation strategy?

## 2. Materials and Methods

### 2.1. Study Area

This study was performed on the Italian island of Asinara (51.9 km²), located northwest of Sardinia (Figure 1). The macrobioclimate of the Island is classified into different bioclimatic types of the thermo- and mesomediterranean thermotypes, from upper dry semihyperoceanic weak to lower subhumid, euoceanic strong [34]. The mean annual temperature is about 17.7 °C, with the highest temperatures in August and the lowest in February. Mean annual precipitation reaches 430 mm, with maximum rainfall between October and April [35]. The bedrock consists mainly of metamorphic complexes (with mica schist, paragneiss, orthogneiss, and migmatite) and intrusive magmatic granite formations [36]. The soil geography is characterized by a mosaic of soil types dominated by variations of leptosols, cambisols, luvisols, and stagnosols [37].

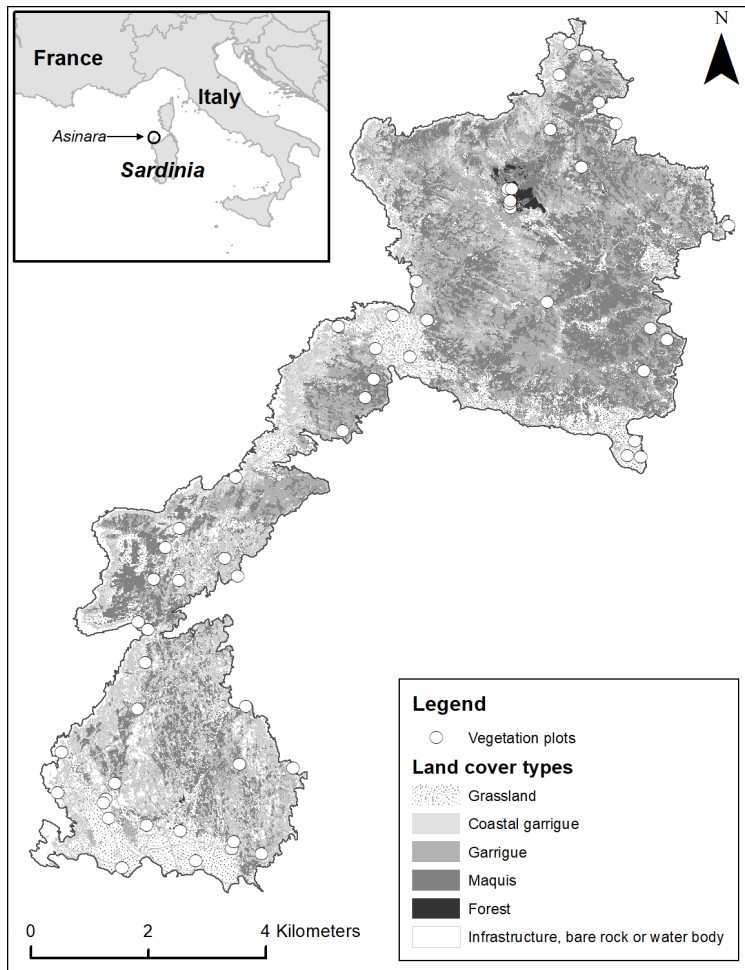

**Figure 1.** Land cover map of the island of Asinara in the northwest of Sardinia (Italy) showing the location of the 57 plots. Land cover types follow Stadtmann et al. [38].

The first human trace on Asinara, a late neolithic tomb, can be dated back to the 4th millennium BC [39]. Until the 19th century, like other Mediterranean islands, Asinara was influenced by a land-use history involving deforestation, farming, terrace cultivation, and animal husbandry [40,41]. Since 1885, the island became a quarantine station, agricultural penal colony, prisoner-of-war camp, and high-security prison [42,43]. During these years, the prison was extended by several branches undertaking different tasks, such as raising cattle or cultivating crops and fodder for livestock [41,43]. Especially in the second half of the 20th century, land use was intensified [44], including the use of fire for clearing land [40]. Asinara stayed a publicly prohibited area for 112 years before

becoming a national park in 1997 [45] and the protected area within the European Natura 2000 network [46]. With the closure of the prison in the same year, agricultural practices were abandoned, and domestic ungulates were released, allowing for their unregulated proliferation [47,48]. Pisanu et al. [49] reported maximum population sizes of 7000 goats, 1000 wild boars, 600 mouflons, 300 donkeys, and 180 horses between 1990 and 2010. Since 2007/2008, goats and wild boars have been captured to reduce their populations. In 2013, approximately 1400 goats, 900 wild boars, 800 mouflons, 400 donkeys, and 150 horses were present on the island [50]. Due to the high reproduction rate of goats, their number has since increased. Because most animals roam free on the island, the density is very heterogeneously distributed, resulting in locally higher grazing pressure [51,52] (Table 2). Consequently, and through processes of succession, the island's vegetation is nowadays dominated by secondary plant communities of garrigue with *Cistus monspeliensis*, open grassland, and semi-open maquis with *Euphorbia dendroides*. Forest formations (*Quercus ilex*, *Pinus pinea*) cover only a minor part of the island [32,38]. A comprehensive floristic description of Asinara was conducted over 30 years ago [53]. Recently, influences of the island's grazing regime on endemic plant species [49], the phytosociological structure [40], as well as the environmental driving factors for differentiation and diversity of the vegetation [51] were examined.

### 2.2. Vegetation and Soil Survey

The used nomenclature for vascular plants follows Conti et al. [54], except for *Olea europaea* var. *europaea* and var. *sylvestris*, which follow Pignatti [55]. Based on a pre-mapping of the island and the aid of orthophotos [38], the main vegetation units, according to the physiognomy of the plant cover, were identified by Drissen et al. [51]. On this basis, 88 plots (10 m × 10 m) were placed randomly, covering these units. Through hierarchical clustering of floristic data, the plots were assigned to different vegetation units (Table 2). The vegetation units were classified into near-natural (primary and climax) and cultural (including secondary successional) conditions. Cultural vegetation units are plant communities that refer to those ecosystems that are managed along a gradient from low-input to high-input systems [13]. For this study, standing vegetation and seed bank were investigated on a subset of 57 plots (each 10 m × 10 m) representing the abovementioned units, as well as a forest stand dominated by *Pinus pinea*. Two plots can be assigned to the maquis scrubland dominated by the cultivar *Olea europaea* var. *europaea*. Because of their anthropogenic origin, both units exemplify the semi-natural character of the island and thus are incorporated into this study.

**Table 2.** Vegetation units identified by Drissen et al. [51]. The number of studied plots (*n*) is indicated. Ungulate abundance is based on point-observation data [51,52]. The units are classified into near-natural (na; primary and climax) or cultural (cu; secondary successional) conditions (following Bocchieri and Filigheddu [32]).

| Abbreviation | *n* | Description | Ungulate Abundance | Condition |
|---|---|---|---|---|
| COA | 11 | Coastal vegetation of primary and secondary garrigues on eroded soils and rocky substrate, mainly at the shoreline and influenced by marine salt-spray, dominated by dwarf scrub, annual forbs, and graminoids. | low | na/cu |
| GRA | 10 | Secondary successional open grassland, widespread at the plain areas in the central and southern part and used for agriculture and raising of livestock during prison times, dominated by annual graminoids, annual forbs, and legumes. | high | cu |

**Table 2.** *Cont.*

| Abbreviation | *n* | Description | Ungulate Abundance | Condition |
|---|---|---|---|---|
| TWG | 6 | Scattered grassland areas with temporarily inundated conditions in winter and spring, mainly in slight depressions in the central and southern part, dominated by graminoids, annual forbs, and legumes. | high | cu |
| CIS | 6 | Widespread secondary xerophytic garrigue scrubland, predominantly in areas of former land use associated with fires for clearing land, dominated by *Cistus monspeliensis*. | high | cu |
| EUP | 10 | Widespread semi-open maquis scrubland dominated by *Euphorbia dendroides* with frequent occurrence of *Pistacia lentiscus*, substitute vegetation for forest formations [32] after fires and destruction [56]. | intermediate | cu |
| JUN | 6 | Climax and secondary maquis scrubland, predominantly located near the coast, characterized by semi-open to preforest formations of *Juniperus phoenicea* subsp. *turbinata*. | intermediate | na/cu |
| PIN | 3 | Fenced afforested stands of *Pinus pinea*. | low | cu |
| QUE | 3 | Fenced holm oak forest remnant of approx. 20 ha at the central northern part, consisting of mature and reforested stands and characterized by a closed canopy of *Quercus ilex* subsp. *ilex*. | low | na |
| OLI | 2 | Large maquis scrubland and preforest formations, mainly located on rocky slopes and in valleys shaped by precipitation-runoff, dominated by the cultivar *Olea europaea* var. *europaea* and annual forbs. | intermediate | cu |

Between March and May 2014, estimated cover-abundance values of all vascular plant species per plot were recorded as a percentage of the total area (10 m × 10 m) using a continuous percentage scale (1–100%) with three divisions for values under 1% (0.1%, 0.5%, 0.7%). To record any late flowering species, the plots were checked again between July and August 2014. We took ten soil cores (depth 3 cm, diameter 4 cm) per plot, five at the end of May, after the emergence period and before seed rain, and five at the end of August 2014, after the seed rain and before the start of the rainy season. Under a Mediterranean climate, the first three centimeters of the topsoil are known to accumulate most of the seed bank [57,58]. The cores were composited to one sample per plot, air-dried, and stored at room temperature until use.

Mediterranean landscapes are characterized by sharp local soil and climate gradients [59]. On the plots, air humidity and air temperature were measured hourly for 12 weeks (mid-April to mid-July 2014) using micro-weather stations (iButton® DS1923, Maxim Integrated, San Jose, CA, USA). During this period, volumetric soil moisture content was measured at regular intervals via time domain reflectometry using five samples per plot (depth 7.5 cm). The plant exploitable soil depth (range 10–115 cm) was recorded with two soil profiles located up- and downhill of each plot. For chemical analyses, composite topsoil samples were collected from five random locations per plot. The pH was measured in 0.01 mol $L^{-1}$ $CaCl_2$ (following DIN ISO 10390). The organic carbon and total nitrogen contents were quantified with a CN elemental analyzer (TruMac, LECO Corporation, Saint Joseph, MO, USA), and the relation of both was calculated. Orthophosphate ($PO_4$-P) was measured photometrically (WTW Photoflex, Xylem Analytics, Weilheim, Germany) (following DIN 38414-4 and ISO 6978).

### 2.3. Seedling Emergence Trial

Between December 2014 and July 2015, a greenhouse germination trial was performed using the seedling emergence method [29,30]. The composite samples were washed through sieves (mesh size 5 mm and 0.2 mm) to increase the soil-seed ratio and promote germination through scarification of the seeds (following ter Heerdt et al. [60]). We checked the fraction larger than 5 mm visually for seeds or bulbs. To test the potential seed availability within

the topsoil samples, we chose to subject each sample to standardized rather than habitat-specific environmental conditions. Each sample was spread in a thin layer (0.3–0.5 mm) on planting trays (20 cm × 16 cm) over sterilized potting mix, placed in the greenhouse, and watered every day. The mean temperature was 19.1 °C (range 11.7–35.6 °C), and the mean relative humidity was 67% (range 25–94%). We used plant luminaries (high-pressure sodium vapor lamp Sirius X400, Bio Green OHG, Bischoffen-Oberweidbach, Germany; 55,000 Lumen at 1.3 m distance) with mean photosynthetically active radiation (PAR) values of 200 µmol m$^{-2}$ s$^{-1}$ (MQ-200, Apogee Instruments, Inc., Logan, UT, USA). The day length of 10 h 22 min was adapted to the mean day length of the study area during vegetation season. Trays were rotated weekly to prevent edge effects. To record any external seed input, control trays were placed randomly. Germination was monitored daily. Seedlings were identified and removed or transferred for further cultivation. When seedling recruitment stagnated after 4 months, the trays were left to dry for 10 days before crumbling the sample layer and starting irrigation again to promote further germination.

*2.4. Data Analyses and Diversity Measures*

Using taxonomical descriptions from Pignatti [55], plant life forms sensu Raunkiær [61] were determined. To identify cultivated plant species, we used the working database of the Italian vascular flora [62]. Seed density was calculated as viable seedlings per m$^{-2}$. Preliminary tests revealed no significant differences between pre- and post-seed-rain samples. Thus, for the final analysis, these samples were combined for each plot. The floristic diversity of each vegetation unit concerning vegetation and seed bank data was evaluated using alpha diversity according to Whittaker [63] (hereafter, species richness) and Simpson's diversity index [64] (hereafter, species diversity). Floristic and environmental data were analyzed using PC-ORD 7.08 (MjM Software, Gleneden Beach, OR, USA). To look for plot-based differences in species composition between vegetation and seed bank, we conducted a Detrended Correspondence Analysis (DCA; [65]) of relative abundances of vegetation cover and seed bank count data, using detrending by segments and down-weighting of rare species. The DCA was overlaid with vectors of plant life forms, weighted by relative species cover. To assess whether the species composition of the seed bank is related to environmental gradients, we performed a Canonical Correspondence Analysis (CCA; [66]). Species response matrix and explanatory matrix with environmental variables (air temperature, percentage of bare soil, C/N ratio, elevation, organic carbon content, pH, phosphate content, relative air humidity, soil depth, slope gradient, soil moisture, and total nitrogen content) were log-transformed. Plant species only found on one plot were excluded (Table S1). Due to collinearity, total nitrogen content was removed from the final ordination. We graphed linear combination scores, and the intraset correlations were used as correlations of explanatory variables and axes [66,67]. The significance of the first axis for both ordination methods was tested by the Monte Carlo permutation test (1000 permutations).

To evaluate if abiotic environmental parameters, plant life form abundances as well as diversity measures and seed density differ significantly between vegetation units, the non-parametric Kruskal–Wallis test ($p \leq 0.05$) followed by multiple pairwise Mann–Whitney U tests ($p \leq 0.05$) with posthoc Bonferroni adjustment were conducted using SPSS Statistics 26 (IBM Corp., Armonk, NY, USA). Concerning parametric variables, a one-way ANOVA with the posthoc Gabriel test ($p \leq 0.05$) was calculated. In the case of inhomogeneous variances, the posthoc Games–Howell test ($p \leq 0.05$) was chosen.

Because *Juncus bufonius* accounted for 40% of all seedlings, the species was removed from the following analyses. Sørensen similarity index [68] was calculated as a similarity measure between the vegetation and the seed bank. To assess the relation between variation in seed bank composition and variation in vegetation composition, a Mantel test [69] was conducted using PC-ORD 7.08 with similarity matrices (rel. Sørensen) for both datasets. Regarding the influence of environmental factors and plant life form abundances on diversity measures, seed density, and species similarity, a stepwise additive multiple regression

analysis was performed using SPSS Statistics 26. Organic carbon content, total nitrogen content, and air temperature were excluded from all final models due to collinearity with the C/N ratio or relative air humidity, respectively.

## 3. Results

### 3.1. Vegetation and Seed Bank Characteristics

On 57 plots, we recorded a total of 361 plant taxa in the vegetation (Table S1). The highest mean species richness per plot was found in the CIS (75.8 ± 3.2) and EUP (70.8 ± 34.7) scrubland, which are classified as cultural vegetation units (Table 2). The near-natural QUE forest (36.0 ± 4.9) and cultural PIN stands (49.7 ± 3.8; Table 3) showed the lowest mean species richness. The highest mean species diversity was found in the near-natural COA (0.91 ± 0.02) and cultural GRA (0.90 ± 0.01) unit, the lowest in the QUE forest (0.52 ± 0.07).

**Table 3.** Species richness and diversity, seedling number, seed density, and Sørensen similarity index for vegetation and soil seed bank in the vegetation units (for abbreviations, see Table 2). Means and standard errors are given. Sørensen index is a measure of similarity between species composition of vegetation and seed bank. Significant differences ($p \leq 0.05$) are marked by lowercase letters. Due to the small number of plots ($n$) for the OLI unit, values are reported without statistical comparison.

| | | Vegetation | | Soil Seed Bank | | | | |
| | $n$ | Species Richness | Species Diversity | Species Richness | Species Diversity | No. of Seedlings | Seed Density per m$^2$ | Sørensen Index |
|---|---|---|---|---|---|---|---|---|
| COA | 11 | 61.64 (±4.58) [a] | 0.91 (±0.02) [a] | 19.00 (±1.76) [ab] | 0.82 (±0.04) [a] | 87 (±12) [a] | 6959 (±921) [a] | 0.37 (±0.03) [ab] |
| GRA | 10 | 59.40 (±3.12) [ab] | 0.90 (±0.01) [ab] | 29.50 (±2.09) [c] | 0.83 (±0.03) [ab] | 355 (±104) [ab] | 28,210 (±8293) [ab] | 0.45 (±0.02) [a] |
| TWG | 6 | 53.00 (±4.91) [ab] | 0.85 (±0.03) [ab] | 27.50 (±1.54) [ac] | 0.40 (±0.07) [b] | 1175 (±236) [b] | 93,490 (±20,900) [b] | 0.44 (±0.03) [a] |
| CIS | 6 | 75.83 (±3.24) [a] | 0.85 (±0.02) [ab] | 24.50 (±2.62) [abc] | 0.89 (±0.02) [a] | 135 (±35) [ab] | 10,703 (±2802) [ab] | 0.37 (±0.01) [ab] |
| EUP | 10 | 70.80 (±4.67) [a] | 0.89 (±0.01) [ab] | 23.40 (±1.66) [abc] | 0.90 (±0.01) [a] | 84 (±13) [a] | 6677 (±1022) [a] | 0.37 (±0.02) [ab] |
| JUN | 6 | 62.33 (±5.45) [ab] | 0.72 (±0.07) [ab] | 16.00 (±1.65) [b] | 0.85 (±0.03) [ab] | 69 (±21) [a] | 5504 (±1672) [a] | 0.30 (±0.02) [bc] |
| PIN | 3 | 49.67 (±3.84) [ab] | 0.65 (±0.07) [b] | 16.67 (±0.33) [ab] | 0.76 (±0.07) [ab] | 84 (±6) [ab] | 6658 (±516) [ab] | 0.23 (±0.01) [b] |
| QUE | 3 | 36.00 (±4.93) [b] | 0.52 (±0.07) [b] | 16.67 (±3.48) [ab] | 0.73 (±0.01) [ab] | 93 (±33) [ab] | 7374 (±2632) [ab] | 0.42 (±0.01) [ac] |
| OLI | 2 | 52.50 (±17.5) | 0.66 (±0.11) | 21.00 (±12.00) | 0.86 (±0.04) | 45 (±27) | 3581 (±2149) | 0.27 (±0.04) |

In the seedling emergence trial, 14,236 seedlings emerged belonging to 213 taxa (Table S1), including four species not documented for Asinara before, which were *Arabidopsis thaliana*, *Silene nocturna*, *Trisetaria michelii*, and *Valerianella locusta*. Six endemic species (sensu Bocchieri and Filigheddu [32]) occurred, *Bellium bellidioides*, *Centaurea horrida*, *Leucojum roseum*, *Limonium acutifolium*, *Romulea requienii*, and *Scrophularia trifoliata*. Mean species richness of the seed bank ranged between 29.5 (±2.1) species in the GRA and 16.0 (±1.7) in the JUN unit, while the highest species diversity was found in the EUP (0.90 ± 0.01) and lowest in the TWG unit (0.40 ± 0.07).

Viable seed density per m$^2$ ranged between 5504 (±1672) in the JUN maquis and 93,490 (±20,900) in the TWG grassland. The latter, however, was influenced by the high abundance of *Juncus bufonius* seedlings, accounting for 79% of the unit's seedlings and 40% of all recorded seedlings, respectively. Excluding *J. bufonius*, the highest seedling density was found in the GRA unit (27,088 ± 8370). Multiple regression showed that seed density (F (4,52) = 61.031, $p \leq 0.001$) is mainly predicted by the seed bank's species richness ($\beta = 0.74$, $p \leq 0.001$) and species diversity ($\beta = -0.52$, $p \leq 0.001$; $r^2 = 0.77$). Seed bank species richness (F (6,50) = 26.377, $p \leq 0.001$) is determined by C/N ratio ($\beta = -0.53$, $p \leq 0.001$), vegetation species richness ($\beta = 0.46$, $p \leq 0.001$) and the percentage of bare soil ($\beta = -0.4$, $p \leq 0.001$), combined reaching $r^2 = 0.61$. Vegetation species richness is also a predictor for seed bank species diversity (F (1,55) = 15.631, $p \leq 0.001$; $\beta = 0.47$, $p \leq 0.01$; $r^2 = 0.21$).

Comparing relative abundances of plant life forms (Table A1), vegetation and seed bank are both dominated by therophytes, accounting for 49.7% (±3.17%) and 84.08% (±2.02%) of the recorded species, respectively. The highest therophyte abundances for both datasets were found in the grassland units (GRA, TWG). Conversely, the other life forms

have higher proportions in the vegetation compared to the seed bank. In particular, this is true for geophytes and phanerophytes, which are almost lacking from the overall seed bank. Figure 2 shows the first two axes of the DCA, representing 12.4% ($p \leq 0.001$) and 14.4% of the variance. The first axis corresponds to a gradient in vegetation structure, separating the seed bank with higher relative abundances of therophytes from the vegetation with higher relative abundances of phanerophytes, nanophanerophytes, and hemicryptophytes.

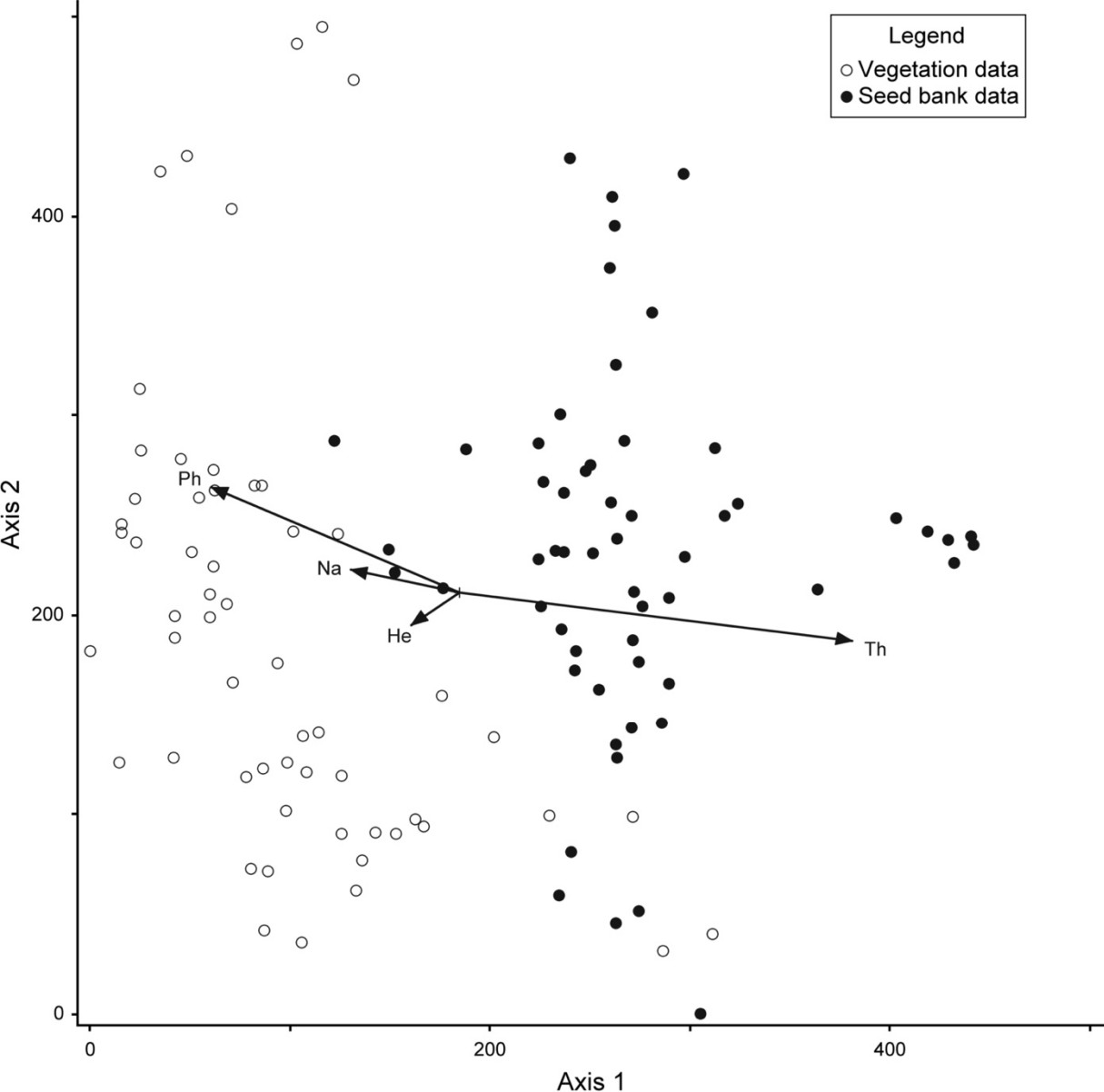

**Figure 2.** First two axes of the DCA ordination of relative species abundance in the vegetation and seed bank graphed as two datasets representing the studied 57 plots. Overlaid are vectors of plant life forms, (cutoff $r^2 = 0.07$). Axis 1: eigenvalue 0.64, gradient length 4.41 SD; axis 2: eigenvalue 0.54, gradient length 4.95 SD; axis 3: eigenvalue 0.36, gradient length 4.42 SD. Axes are scaled in 1 SD = 100. Vectors are scaled to 130% to improve visibility. He = Hemicryptophytes; Na = Nanophanerophytes; Ph = Phanerophytes; Th = Therophytes.

The highest percentage of cultivated plant species in the vegetation was found in the grassland units GRA (13.72 ± 0.83%) and TWG (12.42 ± 0.61%), as well as in the PIN (11.45 ± 0.24%) and OLI (10.71 ± 0.71%) unit (Table 4). Regarding the seed bank, a similar

observation was made for the GRA (10.91 ± 1.14%) and PIN unit (10.05 ± 2.09%), but also for the QUE forest (10.1 ± 1.21%). The lowest percentage of cultivated species was recorded in both vegetation and seed bank of the EUP and CIS scrubland.

**Table 4.** Percentage of cultivated plant species in the vegetation and soil seed bank of the vegetation units (for abbreviations, see Table 2). Means and standard errors are given. Significant differences ($p \leq 0.05$) are marked by lowercase letters. Due to the small number of plots (*n*) for the OLI unit, values are reported without statistical comparison.

|  | *n* | Vegetation | Seed Bank |
|---|---|---|---|
| COA | 11 | 9.89 (±0.82) [ac] | 9.74 (±1.56) [a] |
| GRA | 10 | 13.72 (±0.83) [b] | 10.91 (±1.14) [a] |
| TWG | 6 | 12.42 (±0.61) [ab] | 8.69 (±2.33) [a] |
| CIS | 6 | 8.45 (±1.02) [ac] | 7.56 (±1.66) [a] |
| EUP | 10 | 8.10 (±0.51) [c] | 7.89 (±1.03) [a] |
| JUN | 6 | 9.14 (±0.8) [ac] | 9.67 (±1.37) [a] |
| PIN | 3 | 11.45 (±0.24) [bc] | 10.05 (±2.09) [a] |
| QUE | 3 | 8.96 (±1.22) [ac] | 10.10 (±1.21) [a] |
| OLI | 2 | 10.71 (±0.71) | 8.59 (±2.53) |

In the COA unit, we found eight endemic plant species, the highest number of all units. Of these species, *Bellium bellidioides*, *Limonium acutifolium*, *Romulea requienii*, and *Centaurea horrida* also emerged during the seed bank trial. *Centaurea horrida* is further classified as endangered on the European Red List [70]. The vegetation exhibited five non-native species (following Celesti-Grapow et al. [71]), *Erigeron canadensis*, *Paspalum distichum*, *Phalaris canariensis*, *Pisum sativum*, and *Oxalis pes-caprae*, while the latter also occurred in the seed bank of the GRA grassland.

*3.2. Interrelations of Vegetation and Seed Bank*

Of the 361 species recorded in the standing vegetation, 53% also emerged in the greenhouse trial, whereas 90% of the seedling species were also found in the standing vegetation. The seed bank exhibited 23 exclusive taxa, including the endemics *Leucojum roseum* and *Scrophularia trifoliata*. The JUN maquis showed the highest percentage of species found exclusively in the vegetation (76.0 ± 1.6%), while the TWG grassland showed the highest percentage of species found exclusively in the seed bank (15.8 ± 3.5%; Figure 3).

The Mantel test revealed a medium-strong positive relationship between the plots regarding species composition of the vegetation and seed bank ($r = 0.55$, $p \leq 0.01$), showing that variations of both datasets are associated. While the mean Sørensen similarity between overall vegetation and seed bank was low (0.37 ± 0.01), the index per plot ranged from 0.19 to 0.53, indicating a low to medium species similarity, depending on the vegetation unit. The highest mean species similarity was calculated for both grassland units, GRA (0.45 ± 0.02) and TWG (0.44 ± 0.03), lowest for the PIN (0.23 ± 0.03) and the JUN unit (0.3 ± 0.02) (Table 3). Regression results showed that Sørensen index ($F_{(5,51)} = 9.591$, $p \leq 0.001$) was mainly influenced by C/N ratio ($\beta = -0.54$, $p \leq 0.001$) and the percentage of bare soil ($\beta = -0.34$, $p \leq 0.01$), combined explaining 31.5% of the variance ($r^2 = 0.315$).

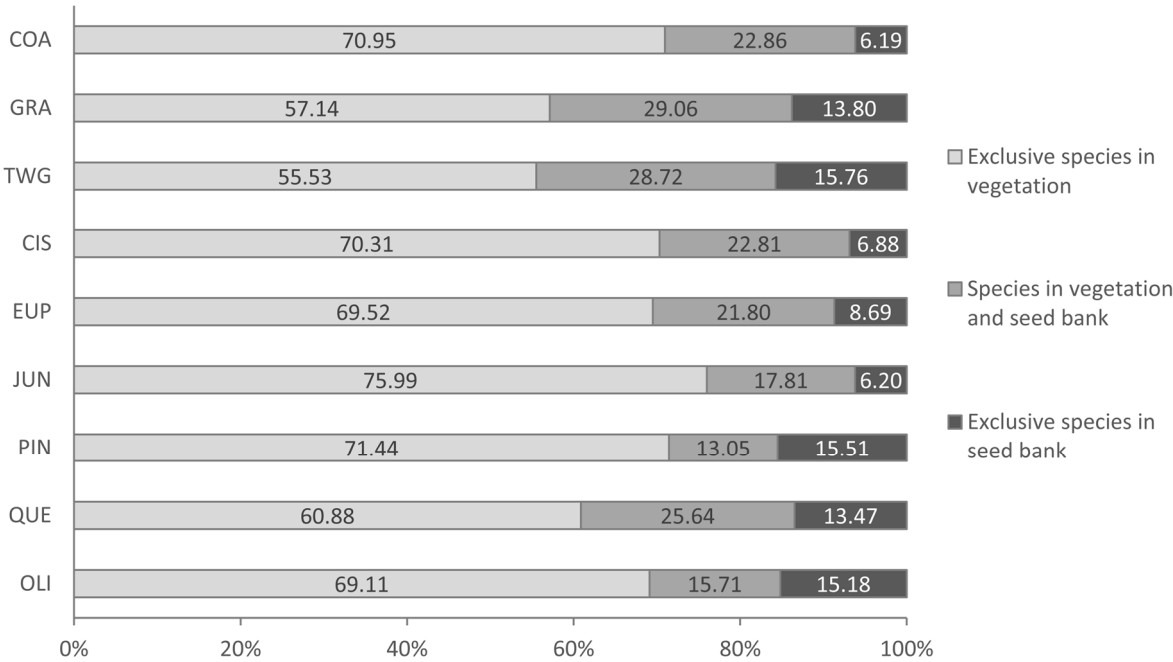

**Figure 3.** Percentage of plant species occurring exclusively in the vegetation, exclusively in the seed bank, and in both datasets for the different vegetation units (for abbreviations, see Table 2). Values are mean percentages of species per plot, rounded to one decimal place. Total number of species per plot (vegetation and seed bank species) are taken as 100%.

### 3.3. Species–Environment Relationships

We found significant differences between the vegetation units concerning all abiotic environmental variables except for pH and phosphate content (Table A2). The CCA, representing a variance (rel. Euclidean distances) of 15.5% ($p \leq 0.001$) for the first and 3.2% for the second axis, revealed the importance of air temperature ($r = -0.89$), C/N ratio ($r = 0.78$), slope gradient ($r = 0.76$), and soil moisture ($r = 0.7$) for the differentiation of seed bank species composition (Figure 4). Elevation, air humidity, and organic carbon content were of secondary importance. Multiple regression revealed positive correlations between slope gradient and C/N ratio ($r = 0.61$, $p \leq 0.01$) and elevation and C/N ratio ($r = 0.55$, $p \leq 0.01$) and negative correlations between air temperature and C/N ratio ($r = -0.66$, $p \leq 0.01$), air temperature and slope ($r = -0.54$, $p \leq 0.01$) and elevation and pH ($r = -0.52$, $p \leq 0.01$).

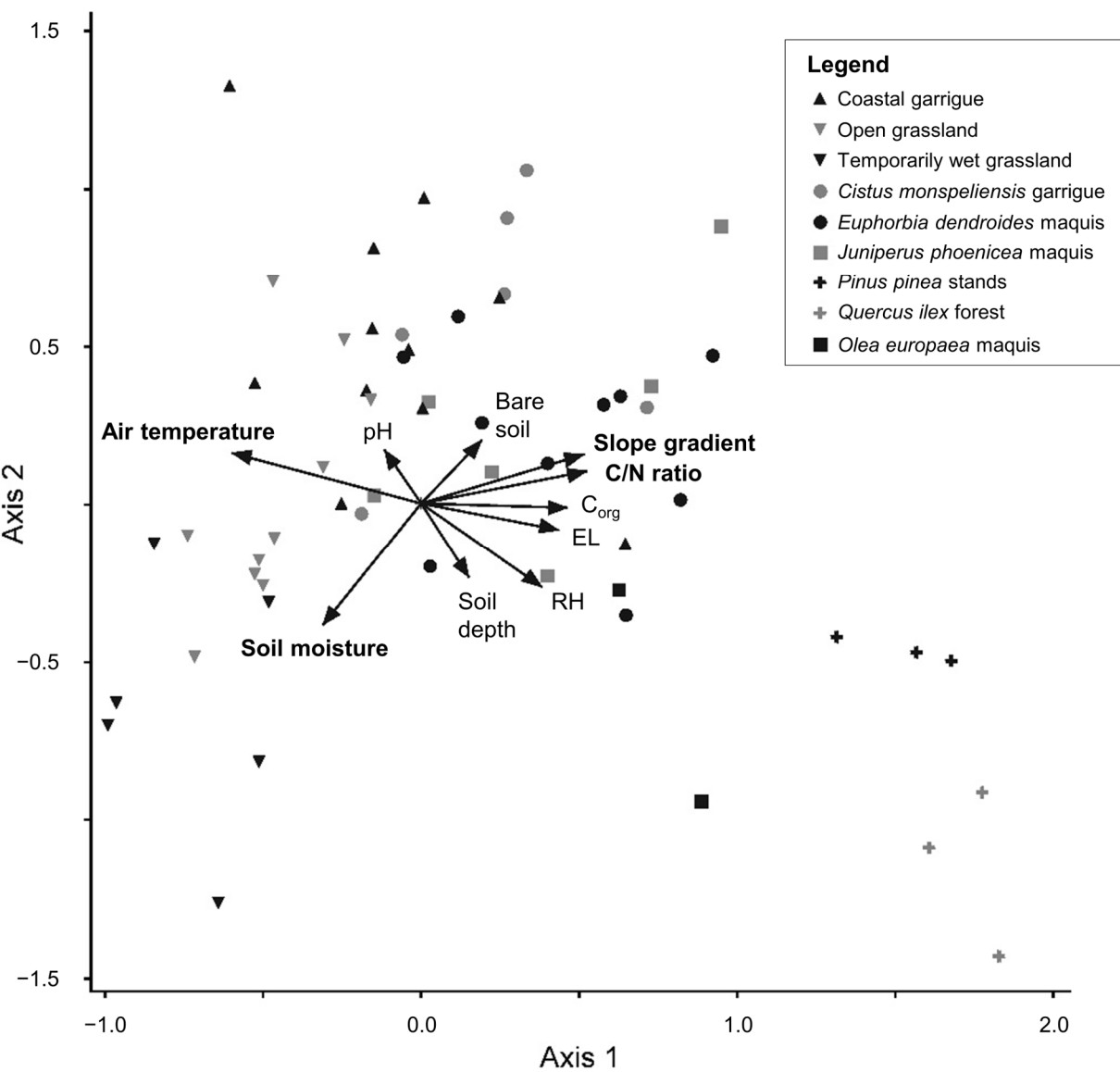

**Figure 4.** First and second axis of the CCA ordination of 57 plots regarding the seed bank (biplot scaling). Points are linear combination scores. Axis 1: eigenvalue 0.45; axis 2: eigenvalue 0.3; axis 3: eigenvalue 0.26. Symbols refer to vegetation units grouped by hierarchical clustering. Environmental variables represented as vectors. Variables that are highly correlated with the first and second axes are given in bold. $C_{org}$ = Organic carbon content; C/N ratio = Ratio of carbon to nitrogen; EL = Elevation; RH = Relative air humidity.

## 4. Discussion

In our study, overall standing vegetation and soil seed bank are dominated by therophytes (Table A1), which is typical for Mediterranean-type climates [72,73]. Therophytes, as short-lived species, form transient seed banks to survive unfavorable conditions such as seasonal drought [27]. We found the highest therophyte abundances for both seed bank and vegetation in the grassland (GRA, TWG). During the trial, germination of phanerophytes was generally scarce. Since phanerophytes often produce larger seeds, which are not easily incorporated into the soil, they are easily subjected to dislocation and seed predation [74–76]. Accordingly, in our samples, seeds of phanerophytes regularly showed feeding traces of insects and rodents. Furthermore, in Mediterranean ecosystems, most woody species and successional shrubs form persistent seed banks [77], which often exhibit mechanisms of innate dormancy [74] that have potentially reduced germination success.

The highest vegetation species richness was found in the semi-open CIS and EUP scrubland (Table 3). Seed bank species richness of these units was intermediate, while the richest seed banks, regarding species number and seed density, were found in the grassland units (GRA, TWG). Grasslands in Europe often show higher values for seed bank species richness compared to other communities [78]. Since floristic richness in the Mediterranean region highly depends on the number of annuals within a community [79], a reason can be found in the grassland's dominating therophytic plant cover. It is not surprising that we found seed bank species richness to be mainly predicted by vegetation species richness but also by a narrower C/N ratio and a higher percentage of bare soil. A narrow C/N ratio favors nutrient availability and plant establishment [80] and, thus, seed bank species richness [57]. The percentage of bare soil represents vegetation gaps that occur due to seasonal plant mortality and disturbance [81]. In these gaps, germination potential is increased, especially of transient seeds [74]. Seed bank species richness itself was found to be the main predictor for seed density.

Since disturbance is an important factor favoring the distribution of light-demanding species and annuals [82], the richest seed banks are often found in the most disturbed habitats [83]. Frequent disturbances such as grazing favor species that invest resources in sexual reproduction [84] and a high number of seeds [72]. Moreover, grazing animals are an important vector for seed movement [85], predominantly through dung [29,86,87]. Seed bank species diversity usually follows a humped distribution with the highest values under intermediate disturbance levels [28], and for Spanish dehesas, Malo et al. [86] found increasing seed bank diversity under grazing influence. Similarly, we found the highest seed bank diversity in the EUP scrubland, where an intermediate abundance of grazing animals was observed [51,52] (Table 2).

Comparing the species diversity of the standing vegetation and the soil seed bank, we found a partly coherent pattern regarding the vegetation units. COA, GRA, CIS, but also the semi-open EUP maquis showed the highest diversity (Table 3). In our trial, *Pistacia lentiscus* and *Euphorbia dendroides*, the dominating species in the standing vegetation of the EUP maquis, had minor germination success. As the species' dominance was not reflected in the seed bank, values for seed bank diversity increased. Consequently, we found vegetation units dominated by large phanerophytes (JUN, OLI, PIN, QUE) to have a higher species diversity in the seed bank compared to the standing vegetation.

Sørensen index showed a low to medium species similarity between vegetation and seed bank. It is known that seed bank and standing vegetation do not always resemble each other [74,77] due to factors such as seed predation [88], vegetative reproduction [28], temporal and spatial segregation [89], or climatic variability [27]. Furthermore, abiotic filters, site history, and interspecific variance in seed production determine differences in species composition [58,90,91]. Since seed-bank components, especially the persistent seed bank, are distributed heterogeneously under the soil surface [92], the number of soil samples and the total sample volume should be high [93] to increase the precision [92] and significance of the findings [93]. Comparing vegetation and seed bank on larger plots with relatively few soil cores and, thus, lower sampling area potentially decreases similarity [76]. Aiming at a high number of replications regarding the vegetation units, the number of soil samples in our study was limited, potentially lowering the significance of our findings. Although a smaller sample size is likely to neglect a significant amount of the rare species, a great loss of information regarding biodiversity is not to be expected [94], and the findings of this study, nevertheless, provide new information on the seed bank as community component, allowing to draw important conclusions [93].

For Mediterranean communities dominated by annuals, a close spatial relationship between vegetation and seed bank is typical [27,72,73], as the seed bank is mainly determined by recent seed rain [95]. Accordingly, in our study, the Sørensen index reached the highest values in vegetation units that are dominated by therophytes, such as the grassland (GRA, TWG). Regarding the lower overall similarity, however, the influence of the high number of feral ungulates on Asinara must be considered. A general pattern of decreas-

ing similarity under grazing [96] is apparent, especially as wild boars are present [97]. However, the similarity is also expected to decrease with the increasing stability of plant communities [78]. Most European forests and perennial grasslands are considered to be long-term stable communities [98], where the seed bank will be composed of long-lived seeds of species from earlier successional stages [99]. Thus, we found the lowest similarity in the PIN stands and OLI maquis, but interestingly, the similarity in the QUE forest was comparably high. Although the dominating holm oak did not emerge during the trial, the germination success of the annual forest understory species was high. It is known that few forest species produce long-lived seeds, as the stable but stressful forest conditions select for species that invest in seedling establishment [100]. Regression results showed that the Sørensen index was mainly predicted by the C/N ratio ($\beta = -0.54$, $p \leq 0.001$) and the percentage of bare soil ($\beta = -0.34$, $p \leq 0.01$). The latter was also mentioned by Ortega et al. [81], who found the abundance of bare ground to be a good predictor for similarity in the Mediterranean pastures of central Spain.

Seed bank composition and density are further affected by abiotic factors, such as topography [81], climate [27,101], nutrient availability [80] and soil moisture [102], as well as biotic factors, e.g., surrounding vegetation [103], disturbances [77,97], and predation [88,104,105]. Regarding our second research question of which environmental factors determine differences in seed bank composition and diversity, we identified microclimatic (air temperature and intercorrelated air humidity), soil physical and chemical (soil moisture, C/N ratio), and topographical (slope gradient) variables to be determinants for species composition of the seed bank (Figure 4). These results are consistent with the findings of Drissen et al. [51] regarding the standing vegetation, showing that the species composition of seed bank and vegetation is determined by the same abiotic factors. Microclimatic conditions affect water availability and soil properties and, thus, influence spatial patterns of seedling emergence [101]. Since the Mediterranean environment is generally moisture-stressed [106], the importance of soil moisture for seed bank composition is not surprising. Several studies highlight the importance of water availability and summer drought, respectively, for plant species distribution and diversity in different Mediterranean ecosystems, e.g., [107–109], and it was found that seed bank species richness increases during wet years [102]. Topographical factors, such as slope gradient, are also important for the abundance and persistence of seeds [81], as well as their spatial distribution in the soil, influenced, e.g., by precipitation run-off [27]. Martinez-Duro et al. [28] found that an increasing slope negatively affects the richness, diversity, and density of the soil seed bank of semi-arid Mediterranean gypsum habitats. Similarly, in the Mediterranean-pluviseasonal climate of Asinara, we found comparably low values for richness, diversity, and density in the units with higher slope gradients, such as the QUE forest and PIN stands. However, the climax stage of these vegetation units, as well as seed predation, must also be considered regarding seed bank characteristics.

Regarding our third research question on indications given by the seed bank for the applied conservation strategy, account must be taken of the fact that the most important aspect of biodiversity conservation in the Mediterranean region is the maintenance of high landscape heterogeneity [110]. Mediterranean habitats have been subjected to disturbances (human, fire, grazing) for a long time, and present phytodiversity largely depends on the continuation of these factors [18]. The disturbance is often necessary to activate the soil seed bank [29], and grazing as a temporarily and spatially predictable disturbance is known to predominantly activate transient seed banks [74]. Nevertheless, to break the dormancy of many persistent seeds, unpredictable disturbances and germination cues such as fire are also needed [74,77]. Several studies have shown that grazing, especially at intermediate levels, promotes floristic richness and diversity [15,18,111]. Besides the activation of the seed bank, the temporal and spatial dispersal of seeds is a key factor in vegetation dynamics [85]. In many open and grazed habitats, herbivorous mammals transport large quantities of seeds, e.g., [86,112], and for Asinara Island, the importance of endozoochorous dispersal was shown for donkeys and goats [87].

On Asinara Island, only small remnants of near-natural vegetation can be found, as most of the island's landscape was shaped for centuries by human influence, resulting in a highly diverse cultural landscape [113]. In the GRA grassland in the south and center of the island, we found the highest number of cultivated plant species in the vegetation and seed bank (Table 4). Since, in the past, these plain areas were used for farming and raising livestock [41], the influence of site history on the floristic composition is illustrated. Similarly, the anthropogenic origin of the PIN stands and OLI maquis is still apparent in today's vegetation and seed bank. Though the CIS and EUP scrubland features fewer cultivated species, these units represent secondary successional stages of natural vegetation, following man-made impacts of fire and overgrazing [32]. Only the QUE forest, JUN maquis, and coastal garrigues (COA) remain as natural or near-natural vegetation units on the island of Asinara (Table 2).

Following the current management plan for the Asinara National Park [114] and the indications for forest restoration given by Mantilla-Contreras et al. [113], the culturally evolved semi-natural vegetation types should be maintained due to their contribution to phytodiversity, while at the same time near-natural oak and juniper forests should be enlarged on at least 20–50% of the island's surface. Grazing animals and wild boars, however, hinder tree regeneration [115] and Mantilla-Contreras et al. [113] ascribed the low regeneration of Phoenician juniper and holm oak on Asinara Island to the influence of the ubiquitous grazing animals. To improve the conservation status of natural habitats and wild plant species of community interest within the European Habitats Directive, the management plan considers controlling the free-roaming domesticated ungulates (mouflons, donkeys, horses) and the eradication of goats and wild boars [114], which is a sensible measure, especially regarding their detrimental influence for forest regeneration [116]. For the coastal garrigues with their endangered endemic elements, a reduction of the numbers of grazing animals or the implementation of low-controlled grazing by sheep [49] should be considered as a viable management option.

In European cultural landscapes, the abandonment of grazing often leads to the loss of phytodiversity [117], and thus, the continuation of (extensive) grazing is an important management tool providing effects of disturbance and creating gaps [118,119]. This is especially of great importance in Sardinia and the entire Mediterranean region, where the decrease in pastures and dwarf-shrub communities due to the abandonment of traditional land-use practices has been identified as a severe problem for nature conservation [31,120]. Accordingly, Dudley et al. [3] argue that it becomes increasingly difficult to maintain pristine areas through non-intervention in an ecological modified landscape, of which Asinara can serve as a prime example. As low naturalness and high floristic diversity coincide in the Asinara National Park, simultaneous preservation of species-rich and diverse culturally evolved habitats such as grassland (GRA, TWG) and semi-open scrubland (CIS, EUP), as well as of near-natural vegetation, such as the highly diverse coastal garrigue (COA), the juniper maquis (JUN) and holm oak forest (QUE) is necessary for an effective protection of floristic diversity. This trade-off between promoting the naturalness of plant communities and maintaining diverse cultural landscapes can be found in various national parks throughout Europe. However, in cases in which, under national law, the implementation of the national park concept varies substantially from the IUCN requirements [5], the designation as a national park should be questioned in the first place.

For the island of Asinara, taking recent studies on the standing vegetation [11,40,49,51] and our findings concerning the soil seed bank into account, we doubt that the requirements of the IUCN national park category of protected areas with its main objective of preserving biodiversity through non-intervention can be met, nor are they adequate regarding the extent of nature development core zones or the planned maintenance of cultural habitats through feral ungulates.

## 5. Conclusions

Based on our results, we suggest an addition to the nature conservation strategy of the Asinara National Park by integrating sustainable land-use management with different degrees of protection as it is practiced in UNESCO biosphere reserves, which can act as a 'real lab' [121]. The biosphere reserve's approach of combining the conservation of biological and cultural diversity corresponds better to the protection of mostly anthropogenic-altered cultural landscapes that are characteristic for regions such as Europe and therewith for protected areas such as the island of Asinara. Additionally, through the integration of local user groups, the people's identification with the island and its environmental value should ultimately be strengthened. In conclusion, we recommend:

- The maintenance of the currently high habitat heterogeneity and plant diversity of the near-natural and culturally evolved vegetation on the island of Asinara by active management;
- Controlled extensive grazing as zoo-anthropogenic disturbance and continuous activation of the soil seed bank;
- The regulation of the grazing animals, particularly the wild boars and goats;
- The expansion and maintenance of fenced grazing-excluded areas to allow the recovery of tree species and sensitive vegetation (e.g., narrow endemics);
- A concept for sustainably integrating the interests of the local community regarding, e.g., science, education, art, and tourism in conservation planning;
- A broad information campaign for visitors to explain and raise public awareness of protection management.

**Supplementary Materials:** The following supporting information can be downloaded at: https://www.mdpi.com/article/10.3390/su142114230/s1, Table S1: Vascular plant species recorded in the standing vegetation (VE) and the soil seed bank (SB).

**Author Contributions:** Conceptualization, S.Z. and J.M.-C.; methodology, C.F., S.Z. and J.M.-C.; software, T.D.; validation, C.F., S.Z. and J.M.-C.; formal analysis, T.D.; investigation, T.D., C.F., J.T.T. and R.S.; resources, S.Z. and J.M.-C.; data curation, T.D., C.F., J.T.T. and R.S.; writing—original draft preparation, T.D.; writing—review and editing, T.D., S.Z. and J.M.-C.; visualization, T.D.; supervision, S.Z. and J.M.-C.; project administration, J.M.-C.; funding acquisition, S.Z. and J.M.-C. All authors have read and agreed to the published version of the manuscript.

**Funding:** This research was funded by the Marianne and Dr. Fritz Walter Fischer Foundation, grant number T192/23337/2012, and the Zempelin Foundation, grant number T214/28727/2016, within the German Stiftungszentrum. We acknowledge financial support by Stiftung Universität Hildesheim.

**Institutional Review Board Statement:** Not applicable.

**Informed Consent Statement:** Not applicable.

**Data Availability Statement:** The data presented in this study are available in the Supplementary Material in Table S1 and on request from the corresponding author.

**Acknowledgments:** We are grateful to the Ente Parco Nazionale dell'Asinara for the support and to Rebecca Winter for field assistance.

**Conflicts of Interest:** The authors declare no conflict of interest.

## Appendix A

**Table A1.** Relative abundances of plant life forms as percentage of standing vegetation and soil seed bank of the vegetation units (for abbreviations, see Table 2). Means and standard errors are given. Significant differences ($p \leq 0.05$) are marked by lowercase letters. Due to the small number of plots (*n*) for the OLI unit, values are reported without statistical comparison.

| | | Chamaephytes | | Geophytes | | Hemicryptophytes | | Nanophanerophytes | | Phanerophytes | | Therophytes | |
|---|---|---|---|---|---|---|---|---|---|---|---|---|---|
| | *n* | Seed Bank | Vegetation | Seed Bank | Vegetation | Seed Bank | Vegetation | Seed Bank | Vegetation | Seed Bank | Vegetation | Seed Bank | Vegetation |
| COA | 11 | 7.09 (±1.52) [a] | 35.02 (±3.90) [a] | 1.51 (±0.54) [a] | 2.87 (±0.51) [ab] | 5.18 (±1.99) [a] | 13.98 (±2.01) [a] | 0.44 (±0.30) [a] | 2.64 (±1.45) [ac] | 0.08 (±0.08) [a] | 4.55 (±1.74) [ac] | 85.71 (±1.65) [a] | 40.94 (±2.76) [ac] |
| GRA | 10 | 3.96 (±3.35) [a] | 3.37 (±2.67) [b] | 0.44 (±0.26) [a] | 1.08 (±0.33) [a] | 4.37 (±1.62) [a] | 14.45 (±2.78) [a] | 0.00 (±0.00) [a] | 0.02 (±0.01) [a] | 0.00 (±0.00) [a] | 0.08 (±0.08) [a] | 91.22 (±3.16) [ab] | 81.01 (±4.11) [b] |
| TWG | 6 | 0.44 (±0.38) [a] | 0.08 (±0.06) [b] | 0.15 (±0.09) [a] | 15.08 (±7.24) [b] | 0.48 (±0.14) [a] | 7.67 (±4.18) [ab] | 0.00 (±0.00) [a] | 0.06 (±0.06) [a] | 0.00 (±0.00) [a] | 0.00 (±0.00) [a] | 98.92 (±0.42) [b] | 77.11 (±7.02) [b] |
| CIS | 6 | 0.46 (±0.46) [a] | 1.12 (±0.60) [b] | 1.51 (±0.60) [a] | 4.22 (±0.80) [ab] | 0.72 (±0.44) [a] | 8.58 (±1.70) [ab] | 9.03 (±2.11) [b] | 35.07 (±4.24) [b] | 0.00 (±0.00) [a] | 5.29 (±2.36) [abc] | 88.28 (±1.63) [ab] | 45.72 (±4.46) [ac] |
| EUP | 10 | 2.46 (±1.76) [a] | 2.94 (±0.78) [ab] | 0.11 (±0.11) [a] | 2.24 (±0.38) [ab] | 3.90 (±1.48) [a] | 9.06 (±1.56) [ab] | 8.29 (±2.03) [b] | 18.24 (±3.56) [bc] | 0.00 (±0.00) [a] | 23.24 (±4.64) [bc] | 85.24 (±3.16) [a] | 44.27 (±5.01) [a] |
| JUN | 6 | 6.33 (±4.15) [a] | 1.83 (±0.66) [ab] | 0.18 (±0.18) [a] | 4.45 (±1.63) [ab] | 12.92 (±6.25) [a] | 6.56 (±1.42) [ab] | 0.00 (±0.00) [a] | 0.80 (±0.29) [ab] | 0.11 (±0.11) [a] | 53.99 (±7.88) [b] | 80.46 (±6.63) [ab] | 32.38 (±6.93) [ac] |
| PIN | 3 | 36.38 (±17.59) [a] | 0.44 (±0.25) [ab] | 0.00 (±0.00) [a] | 3.95 (±1.90) [ab] | 4.26 (±2.61) [a] | 1.21 (±0.44) [b] | 0.69 (±0.69) [ab] | 0.5 (±0.05) [ab] | 1.65 (±1.65) [a] | 55.51 (±8.29) [bc] | 57.02 (±17.81) [a] | 38.38 (±8.59) [ac] |
| QUE | 3 | 31.43 (±14.62) [a] | 0.42 (±0.05) [ab] | 4.48 (±4.16) [a] | 11.71 (±6.92) [b] | 4.58 (±3.01) [a] | 5.67 (±1.75) [ab] | 3.51 (±1.79) [ab] | 0.16 (±0.11) [ab] | 0.00 (±0.00) [a] | 67.12 (±5.86) [bc] | 56.00 (±10.38) [a] | 14.92 (±5.31) [c] |
| OLI | 2 | 1.39 (±1.39) | 0.72 (±0.72) | 2.28 (±2.28) | 3.93 (±1.91) | 25.00 (±8.33) | 5.62 (±0.36) | 0.69 (±0.69) | 0.22 (±0.14) | 0.00 (±0.00) | 70.08 (±8.80) | 70.14 (±3.47) | 19.43 (±6.38) |
| All plots | 57 | 6.88 (±1.81) | 8.25 (±1.96) | 0.91 (±0.27) | 4.6 (±0.98) | 5.28 (±1.08) | 9.78 (±0.95) | 2.73 (±0.64) | 7.54 (±1.72) | 0.11 (±0.09) | 20.12 (±3.44) | 84.08 (±2.02) | 49.7 (±3.17) |

**Table A2.** Abiotic environmental parameters of the vegetation units (for abbreviations, see Table 2). Means and standard errors are given. Soil parameters are meas-ured in topsoil. For microclimate (air temperature, RH), means of daily average values from mid-April to mid-July 2014 are calculated based on a subset of plots. For the same period, means of soil moisture measurements are given. Significant differences ($p \leq 0.05$) are marked by lowercase letters. Due to the small number of plots (*n*) for the OLI unit, values are reported without statistical comparison. C/N ratio = Ratio of carbon to nitrogen; Corg = Organic carbon content; $PO_4$-P = Orthophosphate; TN = Total nitrogen content; RH = Relative air humidity.

| | *n* | Bare Soil (%) | $C_{org}$ (%) | C/N Ratio | Elevation (m) | pH | $PO_4$-P (mg/kg) | Slope Gradient (°) | Soil Depth (cm) | Soil Moisture (vol%) | TN (%) | *n* | Air Temp. (°C) | RH (%) |
|---|---|---|---|---|---|---|---|---|---|---|---|---|---|---|
| COA | 11 | 11.6 (±2.6) [ab] | 3.4 (±0.5) [ab] | 12.9 (±0.3) [ab] | 31.8 (±9.8) [ab] | 5.8 (±0.1) [a] | 1.7 (±0.4) [a] | 12.4 (±2.0) [ac] | 23.7 (±3.6) [ab] | 9.1 (±2.9) [a] | 0.26 (±0.03) [ab] | 11 | 21.4 (±0.2) [ab] | 74.1 (±0.6) [a] |
| GRA | 10 | 5.1 (±1.4) [a] | 1.9 (±0.3) [a] | 10.3 (±0.4) [b] | 14.9 (±2.4) [a] | 6.2 (±0.2) [a] | 2.4 (±0.6) [a] | 4.6 (±0.9) [ab] | 29.2 (±2.8) [ab] | 12.7 (±2.3) [ab] | 0.18 (±0.03) [a] | 5 | 21.2 (±0.1) [ab] | 76.0 (±0.4) [ab] |
| TWG | 6 | 3.7 (±0.8) [a] | 3.2 (±0.6) [abc] | 11.1 (±0.2) [bc] | 29.7 (±14.2) [ab] | 5.8 (±0.4) [a] | 1.0 (±0.2) [a] | 1.7 (±0.4) [b] | 41.8 (±14.5) [ab] | 47.7 (±11.0) [b] | 0.29 (±0.05) [ab] | 3 | 21.6 (±0.1) [a] | 74.9 (±0.3) [ab] |
| CIS | 6 | 16.2 (±2.6) [b] | 3.8 (±0.4) [abc] | 15.1 (±0.7) [a] | 90.3 (±46.2) [ab] | 5.8 (±0.2) [a] | 2.1 (±0.7) [a] | 9.0 (±3.6) [bc] | 20.7 (±1.4) [a] | 6.2 (±1.2) [a] | 0.25 (±0.02) [ab] | 6 | 20.8 (±0.3) [ac] | 75.0 (±0.6) [ab] |
| EUP | 10 | 5.3 (±0.7) [ab] | 4.9 (±0.3) [b] | 14.5 (±0.6) [a] | 45.3 (±10.7) [ab] | 5.9 (±0.2) [a] | 2.0 (±0.4) [a] | 13.1 (±1.8) [ac] | 40.8 (±5.3) [b] | 6.2 (±0.7) [a] | 0.34 (±0.02) [b] | 9 | 20.6 (±0.2) [bc] | 75.1 (±0.4) [ab] |
| JUN | 6 | 16.5 (±7.3) [ab] | 5.2 (±0.8) [ab] | 15.3 (±0.8) [ac] | 20.2 (±6.6) [ab] | 6.3 (±0.4) [a] | 2.6 (±1.0) [a] | 11.5 (±3.9) [bc] | 34.8 (±5.0) [ab] | 10.6 (±2.0) [ab] | 0.34 (±0.04) [b] | 5 | 20.5 (±0.2) [bc] | 77.4 (±0.4) [b] |
| PIN | 3 | 2.7 (±0.7) [ab] | 5.3 (±0.1) [bc] | 21.4 (±0.7) [a] | 268.7 (±7.8) [b] | 5.3 (±0.1) [a] | 5.8 (±1.9) [a] | 17.3 (±0.9) [ac] | 44.5 (±10.3) [ab] | 9.2 (±0.4) [ab] | 0.26 (±0.02) [ab] | 3 | 18.2 (±0.03) [bc] | 81.0 (±0.9) [b] |
| QUE | 3 | 15.0 (±7.6) [ab] | 6.8 (±1.7) [ab] | 15.5 (±1.3) [ac] | 272.3 (±10.9) [b] | 5.2 (±0.2) [a] | 1.6 (±0.2) [a] | 34.0 (±2.5) [c] | 44.8 (±11.4) [ab] | 14.7 (±1.5) [ab] | 0.43 (±0.07) [b] | 3 | 17.6 (±0.1) [c] | 81.1 (±0.6) [b] |
| OLI | 2 | 21.0 (±14.0) | 6.6 (±0.3) | 13.5 (±0.3) | 56.5 (±5.5) | 5.8 (±0.6) | 2.2 (±0.6) | 9.5 (±2.5) | 62.5 (±16.5) | 11.6 (±3.2) | 0.49 (±0.01) | 2 | 19.7 (±0.3) | 79.1 (±2.3) |

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
