# Peer review of "National Park or Cultural Landscape Preservation? What the Soil Seed Bank Reveals for Plant Diversity Conservation"

_sustainability, doi:10.3390/su142114230_

Round 1

Reviewer 1 Report

The paper describes the state of biodiversity in the Asinara National Park in Italy, suggesting a better way of management through the tool of the Unesco biosphere program. The hypothesis is suggestive and certainly the UNESCO biosphere reserves are an approach that allows you to manage in a useful way the traditional landscapes. However this idea clashes with some features of biodiversity management in Italy. In particular, the Italian national parks are managed through a zoning that allows, or at least should allow, to effectively manage the traditional activities, including production, that take place within the park area. These include tourism and fishing of particular importance in the Asinara area. On the other hand, biosphere reserves often, at least within the Italian jurisdiction, do not allow adequate governance. Very often this type of area is in fact included within the areas of the National Park. Therefore, we can conclude that the hypothesis carried out within this papr is not based on a careful analysis of governance and is therefore devoid of application possibilities and adequate feedback.

Author Response

Thank you very much for your comments. Although national parks and biosphere reserves share certain structural characteristics, such as internal zoning, they have different objectives with different management options. We doubt, that Italian National Parks are managed comparably to biosphere reserves. However, some national parks in Europe and e.g., the National Park of Sila in Calabria are also part of or partly designated as biosphere reserves, which certainly can be an option. With our recommendations we do not aim to terminate the national park concept. We rather propose adding certain aspects which are inherent of the biosphere reserve concept, e.g., including local communities, to the nature conservation strategy of national parks with predominantly culturally evolved landscapes.

It is interesting for us to learn that biosphere reserves, within the Italian jurisdiction, often do not allow adequate governance. Acknowledging the difficulties regarding governance is important for practical application. However, our manuscript deals with this issue from a scientific point of view and gives recommendations to improve current practices. We are hopeful that difficulties in governance nevertheless can be resolved based on adequate results.

Reviewer 2 Report

Overall review

I have reviewed the manuscript (ID: sustainability-1725490), “National Park or cultural landscape preservation? What the soil seed bank reveals for biodiversity conservation” submitted to the journal of Sustainability. In this study, the authors were examining the national parks framework in the context of cessation of traditional land-use practices and determining if main conservation goal of afforestation and nature development promotion were maintained within an Italian national park. Specifically, they examined the characteristics of the soil seedbank, standing vegetation, and various environmental factors across different natural and historically disturbed vegetation types. As expected, the comparison between the seed bank and vegetation shows differences, with their seed bank analysis indicating some differentiation across the various natural and cultural vegetation units. Overall, this manuscript is clearly written and raises an important discussion regarding improving the management of disturbed landscapes, which seems to fit within the scope of the journal of Sustainability. However, to strengthen the author’s overall message, stronger links between the premise of the study (i.e. disturbance cessation and management implications) and the experimental design are needed. Listed below are recommendations for a few major and minor revisions:

Major revisions:

-          L394-399 - Be more transparent regarding the limitations of your seed bank – vegetation comparisons given that the sample area examined for the soil seed bank was very small relative to the surveyed area of the aboveground vegetation. See: Plue, J., & Hermy, M. (2012). Consistent seed bank spatial structure across semi‐natural habitats determines plot sampling. Journal of Vegetation Science23(3), 505-516.

-          L145-147 – Provide better context regarding the disturbance history of your study site. For example, within the table or the supplementary info some of these variables should be outlined to provide better context regarding the differences in the vegetation units and the potential reasons as to why they may have different seedbank characteristics. Which vegetation units are currently under more pressure from herbivores? Or is the pressure ubiquitous?

Minor revisions:

-          L17-18/21-22/99 – provide a clear definition of “traditional cultural landscape element”

-          L25 – provide a clear definition of “cultural vegetation units”

-          L65-66 – Emphasize the meaning of this sentence. For example, indicate that the cessation of cultural practices/human disturbances would lead the ecosystem shifting to a degraded woody state with impaired functioning.

-          L86-87 Suggest providing a clear example of what the role of the seed bank is

-          Several minor details regarding your methods are missing:

o   L131-134 Did the cessation of cultural practices end in 1997 or sometime within the 112 years prior?

o   L135-136 over what period of time were these culls conducted? Did they have an impact on abundances?

o   L1140-143/54-155 Why were these vegetation units chosen for your study?  

o   L155-156 Did you use the original 10x10m plots or were they subplots?

o   L164-165 Were cover estimates measured over the 10x10m area or within subplots? How many times did you conduct the surveys per plot? What survey method did you use?  

o   L185-191 – by sample are you referring to each of the soil cores? Or did you bulk each of the five cores from each plot? Were Pre/Post seed rain cores kept separate?

-          L253-256 – what were the differences for diversity and richness for just the vegetation condition? i.e. natural vs cultural

-          L274-293 Consistently refer to the vegetation units, either use the abbreviations or just write out the names

Author Response

Please find here a detailed description on all changes made:

-          L394-399 - Be more transparent regarding the limitations of your seed bank – vegetation comparisons given that the sample area examined for the soil seed bank was very small relative to the surveyed area of the aboveground vegetation. See: Plue, J., & Hermy, M. (2012). Consistent seed bank spatial structure across semi‐natural habitats determines plot sampling. Journal of Vegetation Science, 23(3), 505-516.

Thank you for this advice. We stated the limitations of our seed bank sampling (lines 526-536).

-          L145-147 – Provide better context regarding the disturbance history of your study site. For example, within the table or the supplementary info some of these variables should be outlined to provide better context regarding the differences in the vegetation units and the potential reasons as to why they may have different seedbank characteristics. Which vegetation units are currently under more pressure from herbivores? Or is the pressure ubiquitous?

We added more context regarding the disturbance history of Asinara island (lines 158-166) and we extended Table 2 regarding information about site history and a categorisation of ungulate abundance.

-          L17-18/21-22/99 – provide a clear definition of “traditional cultural landscape element”

We added a definition in the text (lines 122-124).

-          L25 – provide a clear definition of “cultural vegetation units”

We added a definition in the text (lines 213-215).

-          L65-66 – Emphasize the meaning of this sentence. For example, indicate that the cessation of cultural practices/human disturbances would lead the ecosystem shifting to a degraded woody state with impaired functioning.

We emphasized this aspect (lines 76-79).

-          L86-87 Suggest providing a clear example of what the role of the seed bank is

We gave an example of the role of the seed bank (lines 101-110).

-          Several minor details regarding your methods are missing:

o   L131-134 Did the cessation of cultural practices end in 1997 or sometime within the 112 years prior?

We clarified this sentence (lines 168-169).

o   L135-136 over what period of time were these culls conducted? Did they have an impact on abundances?

The numbers of goats and wild boar were reduced. We clarified this issue based on the available information (lines 170-175).

o   L1140-143/54-155 Why were these vegetation units chosen for your study?

We clarified this issue (lines 207-211).

o   L155-156 Did you use the original 10x10m plots or were they subplots?

We used the original plots. We clarified this issue (line 216).

o   L164-165 Were cover estimates measured over the 10x10m area or within subplots? How many times did you conduct the surveys per plot? What survey method did you use? 

The cover estimates were measured over the entire 10x10 m area, which is now stated clearer in the text (line 216, 225-226).

The surveys were conducted twice per plot, we added this information in the text (lines 244-245).

The estimated cover-abundance values of all vascular plant species per plot were recorded as percentage of the total area (10 m x 10 m) using a continuous percentage scale (1 % - 100 %) with three divisions for values under 1 % (0.1 %, 0.5 %, 0.7 %). We included this information in the text (lines 226-244).

o   L185-191 – by sample are you referring to each of the soil cores? Or did you bulk each of the five cores from each plot? Were Pre/Post seed rain cores kept separate?

By sample we mean the composite sample of five cores per plot per season (line 245-247). We kept the pre- and post-seed rain samples (each consisting of five cores) separate during the seedling emergence trial. The analyses did neither yield significant differences regarding the comparison of the units divided by sampling-season nor did they show clear patterns, so that for the final analysis we combined the pre- and post-seed rain sample results (line 249). We added this information in the text (lines 288-289).

-          L253-256 – what were the differences for diversity and richness for just the vegetation condition? i.e. natural vs cultural

We rephrased the description regarding the comparison of natural and cultural vegetation units. Values for vegetation diversity and richness can be found in Table 3 and in lines 333-338, 489-490.

-          L274-293 Consistently refer to the vegetation units, either use the abbreviations or just write out the names

We now predominantly use the abbreviations for the vegetation units. However, to aid the reading of the text, we sometimes use both, name and abbreviation.

Reviewer 3 Report

In this manuscript, the authors studied the soil seed bank for biodiversity conservation of a semi-natural land in Italy. The aim of the study is in line with the scope of this journal and the research is of interest for its semi-natural land-uses conservation and restoration and sustainable land-use management information. The article is well prepared and the results are clearly presented, illustrated, and discussed. The conclusions or summary are accurate and supported by the content. I have some recommendations for authors:

Line 2-3: Please use uppercase for each word in the title

Line 16:  important role in

Line 22: and the main 

Line 24: were

Line 29: applies to

Line 35: for humankind

Line 42: except for research

Line 50: “basically” may be an unnecessary word in this sentence 

Line 86: decades,

Line 112: Study Area

Line 114: classified into different 

Line 116: Please double-check the spelling it should be “ oceanic”?

Line 116: The mean

Line 119 and 137: Please double-check the formatting, you may delete the unpubl. data from the text and just referenced to the reference list

Line 128: becoming a quarantine station

Line 132:  and the protected area

Line 160:  incorporated into this study. 

Line 286: Figure 2. should be bold

Line 185: Out of my curiosity any reason has the work didn’t get published so far? 

Line 309: 3.2. Interrelations of Vegetation and Seed Bank

Line 332: 3.3. Species-environment Relationships

Line 449: The disturbance

Line 450: as a temporarily 

Line 486:  landscapes,

Line 493: species-rich

line 522: of protection management. 

Author Response

Thank you very much for your comments, please find a description of our changes below:

Line 2-3: Please use uppercase for each word in the title

Line 16:  important role in

Line 22: and the main

Line 24: were

Line 29: applies to

Line 35: for humankind

Line 42: except for research

Line 50: “basically” may be an unnecessary word in this sentence

Line 86: decades,

Line 112: Study Area

Line 114: classified into different

The errors mentioned above were corrected (lines 2-139).

Line 116: Please double-check the spelling it should be “oceanic”?

This is not a spelling error. „Euoceanic“ describes a certain oceanic macrobioclimate in the Mediterranean with a low continentality index value (line 141).

Line 116: The mean

This was corrected (line 141).

Line 119 and 137: Please double-check the formatting, you may delete the unpubl. data from the text and just referenced to the reference list

We included the unpublished data in the reference list (lines 771-772, 804-805) and adjusted the reference numbering accordingly.

Line 128: becoming a quarantine station

Line 132:  and the protected area

Line 160:  incorporated into this study.

Line 286: Figure 2. should be bold

The errors mentioned above were corrected (lines 161-394).

Line 185: Out of my curiosity any reason has the work didn’t get published so far?

The work hasn’t been submitted anywhere for publication so far. It is part of a dissertation, and the submission was delayed due to personal circumstances of the first author.

Line 309: 3.2. Interrelations of Vegetation and Seed Bank

Line 332: 3.3. Species-environment Relationships

Line 449: The disturbance

Line 450: as a temporarily

Line 486:  landscapes,

Line 493: species-rich

line 522: of protection management.

The errors mentioned above were corrected (lines 426-683).

Reviewer 4 Report

The manuscript starts from the paradoxical point of high biodiversity in the cultural landscape and low biodiversity without human impact in the Mediterranean region. Taking Asinara National Park as an example, the manuscript investigates the diversity of the seed bank and standing vegetation in different habitats, as well as the analysis of environmental influences, and finally proposes a strategy for biodiversity and sustainable development. Overall, the manuscript is well-structured, and the scientific issues are clear and interesting. But improvements are still needed and some specific suggestions and questions are listed below.

The experimental results of the manuscript provide evidence of plant diversity only, should the title be changed from "biodiversity" to "plant biodiversity"?

L47, it is proposed to add to Table 1 a comparison of the different types of PAs in the Italian Framework Law on PAs in terms of definition, objectives and management, in particular the category of national parks, and thus with the IUCN categories.

L113, what are the hypothesis of the questions?

L242-247, why did the DCA and CCA methods chosen in this manuscript and not other methods, such as Detrended Canonical Correspondence Analysis (DCCA)?

L248-251, why were these environmental variables chosen? How did the experiment control for environmental variables in the samples when exploring the characteristics of the seed bank and standing vegetation in the different habitats explored?

L560-571, the beginning of the sentence should be capitalized.  

Author Response

The experimental results of the manuscript provide evidence of plant diversity only, should the title be changed from "biodiversity" to "plant biodiversity"?

We have changed the title to “plant diversity” and three passages in the text (lines 28, 139, 709), as well as the supplementary material, accordingly.

L47, it is proposed to add to Table 1 a comparison of the different types of PAs in the Italian Framework Law on PAs in terms of definition, objectives and management, in particular the category of national parks, and thus with the IUCN categories.

With Table 1 global categories of protected areas are presented. For an implementation of one of these categories, a designation under national law of a country is unquestionably necessary. However, adding PAs that are derived under national law to this table would disrupt the consistency of the information presented, since those national PAs would have to be assigned to equivalent international category, rather than added to the list.

As we investigated Asinara National Park as a case study for the management of ecological modified landscapes, our conclusions apply to many other national parks throughout Europe. Thus, in this context, we would like to politely abstain from adding Italian national PAs to Table 1.

L113, what are the hypothesis of the questions?

To reduce redundancy, we chose to only include the research questions and not both, hypotheses and questions. However, if desired, we can add the hypotheses to the research questions.

(1) We hypothesised that seed bank and standing vegetation differ significantly, overall and habitat-specific, in terms of species composition, species richness and diversity, plant life forms, and occurrence of cultivars.

(2) We hypothesised that seed bank species composition and diversity are driven by environmental factors, namely soil physical and chemical, topographical, and climatic gradients, which might be similar to the environmental factors that determine species composition and diversity of the standing vegetation.

(3) We hypothesised that species richness and diversity of the seed bank is higher in semi-natural habitats with a higher degree of disturbance, and that a conservation strategy that targets biodiversity has to account for the maintenance of those habitats.

L242-247, why did the DCA and CCA methods chosen in this manuscript and not other methods, such as Detrended Canonical Correspondence Analysis (DCCA)?

As indirect gradient analysis such as DCA and direct gradient analysis such as CCA serve different purposes in ecological analyses, we performed both techniques. As we were firstly interested in how the species composition differs between standing vegetation and soil seed bank, we performed a DCA, which performed well on our dataset. To further assess, whether the species composition of the seed bank is related to environmental gradients, we performed a CCA. We clarified this information to the text (lines 310, 314-315).

A NMDS conducted with our dataset resulted in a less clear image, which is the reason this was discarded. A DCCA generally makes sense, if many and redundant variables are part of the analysis (Leyer & Wesche 2007). Since we left out collinear variables and the number of variables was low compared to the number of observations, the CCA results were credible. Furthermore, as the CCA did not show an arch effect, detrending and rescaling was not necessary.

L248-251, why were these environmental variables chosen?

How did the experiment control for environmental variables in the samples when exploring the characteristics of the seed bank and standing vegetation in the different habitats explored?

The environmental variables were chosen because they represent important site properties. Since Mediterranean landscapes are characterised by sharp local soil and climate gradients (Aurelle et al. 2022), abiotic environmental factors (e.g., topography, climate, soil physical and chemical parameters) are likely to have an impact on plant establishment, floristic patterns and species distribution of vegetation and seed bank [e.g. references 16, 18]. We added part of this information as well as a recent reference to the text (lines 256-257).

To test the potential seed availability within the top-soil samples, we chose to subject each sample to standardised rather than habitat-specific environmental conditions regarding lightning, microclimate, soil moisture and substrate properties. We added this information to the text (lines 275-277).

L560-571, the beginning of the sentence should be capitalized.

Done.

Reviewer 5 Report

The authors investigated the soil seed bank, standing vegetation, and environmental factors in different cultural and natural habitats. Since the highest species richness and diversity was were revealed for cultural vegetation units, they need to be of primary concern regarding the preservation of the island’s phytodiversity. This manuscript is well organized and the drawn conclusions are coherent with the obtained results. Given the main objective of the conservation of biodiversity in the Asinara National Park, we conclude that a biosphere reserve with an adapted sustainable land-use management might be more suitable than a national park to maintain the high biodiversity of the island. This conclusion applies for to many other national parks throughout Europe.

Lines 34 – 36: I think that you should add this recent references as example to support your sentence “Despite numerous efforts regarding biodiversity conservation, its loss continues to be a global problem of high significance for the humankind, with far-reaching ecological and socio-economic consequences [1,2].”. I would like to suggest:

Ahmad, F., et al. (2022). Patterns of spatial distribution, diel activity and human-bear conflict of Ursus thibetanus in the Hindu Kush mountains, Pakistan. Global Ecology and Conservation, 37, e02145.

Lines 354 – 356: Please, add all the acronyms on the y-axis in the caption.

Author Response

Lines 34 – 36: I think that you should add this recent references as example to support your sentence “Despite numerous efforts regarding biodiversity conservation, its loss continues to be a global problem of high significance for the humankind, with far-reaching ecological and socio-economic consequences [1,2].”. I would like to suggest:

Ahmad, F., et al. (2022). Patterns of spatial distribution, diel activity and human-bear conflict of Ursus thibetanus in the Hindu Kush mountains, Pakistan. Global Ecology and Conservation, 37, e02145.

Thank you for the recommended publication. With the first paragraph of the introduction, we want to describe biodiversity loss as a global problem, rather than referencing case studies at this point. Thus, we would like to politely abstain from adding the suggested publication.

Lines 354 – 356: Please, add all the acronyms on the y-axis in the caption.

The caption of Figure 3 was missing a reference to Table 2, where the acronyms are explained. We added ‘(for abbreviations see Table 2)’ as we did for the other figures and tables (line 476).

Round 2

Reviewer 1 Report

The authors did not actually respond to my criticisms. the problem is that the basic assumption (the management of a national park with a UNESCO approach) makes no real sense. Hence the statement "our manuscript deals with this issue from a scientific point of view and gives recommendations to improve current practices" is itself meaningless. The article should be rewritten on completely different assumptions.

Author Response

Dear reviewer,

thank you for your response. From our point of view our assumption (the management of a national park with a UNESCO approach) makes sense as explained before. In addition, we do not want to adress specific problems of Italy.

We have chosen Asinara as a case study for our investigation and yes, Asinara is an Italian National Park. But those problems to which we refer do arise in several other European Nationalparks,  e.g. "Unteres Odertal" and "Jasmund" in Germany, "Gorce National Park" in Poland, "Cabañeros National Park" in Spain.